# Nanoporous Crystalline Composite Aerogels with Reduced Graphene Oxide

**DOI:** 10.3390/molecules25225241

**Published:** 2020-11-10

**Authors:** Christophe Daniel, Baku Nagendra, Maria Rosaria Acocella, Esther Cascone, Gaetano Guerra

**Affiliations:** Dipartimento di Chimica e Biologia “Adolfo Zambelli”, Università degli Studi di Salerno, Via Giovanni Paolo II 132, 84084 Fisciano, Italy; nbaku@unisa.it (B.N.); macocella@unisa.it (M.R.A.); esthercascone@hotmail.it (E.C.); gguerra@unisa.it (G.G.)

**Keywords:** aerogels, sol-gel process, supercritical carbon dioxide extraction, syndiotactic polystyrene, poly(2,6-dimethyl-1,4-phenylene oxide), reduced graphene oxide, catalytic oxidation

## Abstract

High-porosity monolithic composite aerogels of syndiotactic polystyrene (sPS) and poly(2,6-dimethyl-1,4-phenylene oxide) (PPO) containing reduced graphene oxide (r-GO) were prepared and characterized. The composite aerogels obtained by supercritical carbon dioxide (scCO_2_) extraction of sPS/r-GO and PPO/r-GO gels were characterized by a fibrillar morphology, which ensured good handling properties. The polymer nanoporous crystalline phases obtained within the aerogels led to high surface areas with values up to 440 m^2^ g^−1^. The role of r-GO in aerogels was studied in terms of catalytic activity by exploring the oxidation capacity of composite PPO and sPS aerogels toward benzyl alcohol in diluted aqueous solutions. The results showed that, unlike sPS/r-GO aerogels, PPO/r-GO aerogels were capable of absorbing benzyl alcohol from the diluted solutions, and that oxidation of c.a. 50% of the sorbed benzyl alcohol molecules into benzoic acid occurred.

## 1. Introduction

Since the first examples of synthetic polymer aerogels reported by Pekala and co-workers [1,2,3], many types of polymer aerogels have been synthetized by changing the sol-gel process and the polymer backbone [4,5,6,7,8,9,10,11,12]. Due to the combination of low cost, simple production processes, and straight-forward tailoring of the physicochemical and functional properties [13,14,15], polymer aerogels continue to attract significant attention from academic and industrial communities for a diverse range of uses within the scope of a broad technological platform [16,17,18,19,20,21,22,23,24].

In particular, syndiotactic polystyrene (sPS) and poly(2,6-dimethyl-1,4-phenylene oxide) (PPO) aerogels endow mixed porosity, namely, nanoporosity (or ultra-microporosity according to IUPAC classification; <0.7 nm) located in the polymer crystalline phase as well as mesoporosity and macroporosity within the three-dimensional (3D) fibrillar surface voids [25,26,27]. The nanoporous crystalline phases of sPS and PPO are able to sorb volatile organic compounds (VOCs) from water and vapour phase even at low activities [28,29,30,31,32,33]. Nanoporous PPO and sPS aerogels present the high sorption capacity typical of their respective nanoporous crystalline phases associated with a fast sorption kinetics due to high mesoporosity and macroporosity [25,26,27]. Therefore, sPS and PPO nanoporous crystalline aerogels are particularly interesting for potential use in air and water purification devices. In addition to this possible use, applications in microelectronics [34] and for the removal of airborne nanoparticles [35,36] have been also proposed for sPS aerogels.

High-porosity polymeric aerogels are also especially appealing for supporting catalysts as they are chemically inert and mechanically stable with high durability and they offer many benefits, such as high surface area and high sorption. Many examples of aerogel catalysts have been reported in literature [37,38,39,40,41].

Graphene oxide (GO) and other carbon-based oxides have long been highly attractive for use as catalysts, catalyst carriers, and catalytic reagents in organic molecule transformations [42,43,44,45,46,47,48,49] due to their unique characteristics, which include enriched chemical structures, functional active sites (acidic and basic), wreckages, defects, unpaired π-electrons of carbon, and larger specific surface areas. Moreover, the addition of carbon-based materials has been also shown to improve the efficiency of metal oxide photocatalysts [50,51]. As the catalytic efficiency strongly depends on active sites and available surface area of the catalyst [43,44,45,49], undesired large polydisperse particles are formed. The prevention of aggregation is also essential to fully exploit the physical properties of GO and obtain composite materials with superior functional properties [50,52,53,54].

In this study, we fabricated monolithic aerogels based on nanoporous crystalline PPO and sPS polymers and reduced graphene oxide (r-GO) hybrid nanocomposites by making use of the advantages of nanoporous crystallinity of the polymers as well as the GO catalytic activity. The uniform dispersion of r-GO was controlled in a two-step approach, gelation followed by supercritical carbon dioxide (scCO_2_) extraction procedure. The monolithic composite sPS/r-GO and PPO/r-GO aerogels were characterized by a fibrillar morphology, which ensured good mechanical properties, and large surface area values up to 440 m^2^ g^−1^ for PPO/r-GO aerogels.

The oxidizing ability of composite aerogels and the possible use of these materials as selective heterogeneous catalytic agents for the transformations of organic pollutants from aqueous solutions were assessed by exploring the oxidation of benzyl alcohol from diluted aqueous solutions, as GO displays a high propensity to oxidize alcohols. Preliminary tests reported here showed that, while sPS/r-GO aerogels did not present any oxidation of benzyl alcohol as a result of the absence of sorption within the aerogels, PPO/r-GO aerogels were capable of absorbing benzyl alcohol from the diluted aqueous solutions and oxidizing a fraction of the sorbed benzyl alcohol molecules.

## 2. Results

### 2.1. Reduced Graphite Oxide (r-GO) Preparation and Characterization

The X-ray diffraction patterns of the starting graphite (G), the resultant graphene oxide (GO) obtained by the Hummers’ method, and the final reduced graphene oxide (r-GO) obtained by solvothermal reduction followed by extraction with scCO_2_ are reported in Figure 1.

The oxidized graphite (GO) formation was confirmed by an increase in the distance between the layers from 0.34 nm up to 0.84 nm, while the *d*-spacing corresponding to the in-plane periodicities of 100 planes remained intact. After solvothermal reduction of GO, we observed the X-ray diffraction pattern, the maintenance of the in-layer graphitic order (100 reflection plans), and the displacement of the peak (*d* = 0.84 nm with a peak at *d* = 0.35 nm), which clearly indicated the formation of r-GO. Another notable difference was in the full-width at half maxima (FWHM) of the r-GO, which increased compared with that of pristine graphite (G), confirming the formation of the disordered graphitic structure after oxidation followed by solvothermal reduction treatments.

### 2.2. Polymer/r-GO Monolithic Aerogel Preparation and Characterization

Schematic illustration (Figure 2A) shows the method for the preparation of homogeneously dispersed monolithic aerogels of polymer/r-GO nanocomposites.

In the first step, the freshly prepared GO was dispersed in 1,4-dichlorobenzene (DCB) solvent (see photo a). Subsequently, solvothermal reduction treatment was performed in a bath sonicator, which promoted the simultaneous reduction of oxygenated functional groups such as epoxy, hydroxyl, and carboxylic groups, as well as the removal of surface H_2_O molecules from the GO structure, leading to the weakening of the interlayer attraction. On further sonication, complete layer separation took place, resulting in an exfoliated and stable dispersion in the hydrophobic solvent (see photo b). Then, polymer was added to this stable dispersion and after complete dissolution of the polymer and obtaining a homogeneous solution by heating, the solution was cooled down to room temperature where the gelation occurred (see photo c). Subsequently, the gel solvent was extracted with supercritical carbon dioxide (scCO_2_), which led to the formation of monolithic aerogels with uniformly dispersed polymer/r-GO nanocomposites, as shown in the final step (see photo d).

ScCO_2_ solvent extraction procedure offered a monolithic aerogel of hybrid polymer/r-GO nanocomposites with retention of both the shape and size of the gel with negligible shrinkage (Figure 2A; photos a and b). It is also worth noting that the monolithic composite aerogels displayed good handling properties as can be seen in Figure 2B.

For monolithic aerogels with a regular shape, the total porosity, including macroporosity, mesoporosity, and microporosity, can be estimated from the volume/mass ratio of the aerogel. The density and total porosity values of neat polymer aerogels and polymer/r-GO aerogels obtained with sPS and PPO are listed in Table 1.

The typical SEM pictures of a pure sPS aerogel and a hybrid sPS/r-GO aerogel are compared in Figure 3.

The SEM pictures (Figure 3A,B) showed that, both for pristine sPS aerogel and the hybrid sPS/r-GO aerogel, an open network of fibrils with a diameter of 50–200 nm was obtained. Most pores of the network were 0.1–0.5 μm wide. We were also able to observe, on the SEM micrograph of the composite aerogel (Figure 3B), the absence of large size r-GO aggregates. In fact, the SEM images displayed the presence of r-GO particles (some of them are indicated by yellow arrows) dispersed in the monolithic polymer aerogels having an average lateral size of c.a. 200–750 nm. The same fibrillar morphology was also observed for PPO-based aerogels (not shown here).

Figure 4 shows the X-ray diffraction patterns of r-GO and hybrid polymer/r-GO nanocomposite monolithic aerogels containing 20 wt.% of r-GO.

The 002 reflection corresponding to r-GO at 2θ = 25.3° was not detected in any of the two nanocomposite aerogels. This absence of the r-GO reflection clearly indicated that most of the GO was constituted by structural layers exhibiting negligible order in the direction perpendicular to the graphitic structure due to homogeneous dispersion of r-GO nanoparticles in the polymer matrix [55].

The X-ray diffraction patterns also showed that, after the guest (solvent) extraction procedure, nanoporous crystalline polymer phases were obtained for both composite aerogels. Curve b of Figure 4 showed diffraction reflections at 2θ = 8.3°, 13.5°, 16.4°, 20.5°, and 23.6°, typical of the known crystalline nanoporous structure, δ-phase [56], for the sPS_0.8_/r-GO_0.2_ monolithic aerogel, while the PPO_0.8_/r-GO_0.2_ aerogel diffraction pattern (curve c of Figure 4) appeared with diffraction reflections at 2θ = 4.5°, 6.9°, and 11.1° corresponding to the α PPO nanoporous crystalline structure modification [57].

Sorption-desorption N_2_ isotherms (where sorption is expressed as cm^3^ of nitrogen in normal conditions per gram of polymer) collected for pristine polymer aerogels and composite aerogels containing 20 wt.% of r-GO are reported in Figure 5.

The shape of isotherms, associated with the presence of hysteresis, suggested type IV isotherm with the presence of big mesopores.

The N_2_ physisorption at 77 K allows to characterize pores in the mesopore and micropore range. The total meso and macropore volume can be customarily estimated from the volume of the liquid N_2_ sorbed at a predetermined P/P_0_ value in the upper part of the isotherm curve. However, the presence of macropores shown by the SEM measurements (Figure 3) associated with large mesopores precluded an accurate determination of the total pore volume since adsorbed gas volumes increased asymptotically as relative pressure approached unity. The total pore volume (V_tot_N_2_) estimated as the volume of liquid N_2_ at P/P_0_ ≈ 0.95 (row 5 of Table 1) differed greatly from the total porosity (i.e., macro-, meso-, and micro-porosity) of the aerogels (row 3 of Table 1). We also noted that the mesoporosity calculated with the BJH method (row 6 of Table 1) represented only a small fraction of the total porosity. This result agreed with those of the SEM micrographs (Figure 3) that showed the strong macroporous nature of all the aerogels.

In addition to the BJH analysis which revealed an average mesopore radius of c.a. 9 nm for all the aerogels with the exception of the sPS_0.8_/r-GO_0.2_ aerogel which displayed an average radius of 6.3 nm (row 7 of Table 1), the presence of microporosity was also verified by the DFT method. The DFT analysis showed that micropores were present in the range of 0.13–0.20 nm and it also appeared that the micropore volumes were about two times larger for PPO aerogels than the values calculated for sPS aerogels (row 8 of Table 1). These larger values can be attributed to the nanoporous nature of both the amorphous and the crystalline phases of PPO samples obtained after solvent removal [30], while for sPS, only the crystalline phase was nanoporous.

The DFT results were consistent with the BET values (row 4 of Table 1). For instance, the measured surface area for the pure PPO aerogel was 550 m^2^ g^−1^, while the BET value obtained for the pure sPS aerogel was 290 m^2^ g^−1^. For both PPO and sPS nanocomposite aerogels, a decrease in the surface area was observed. In first approximation, this decrease could be attributed to the r-GO fraction as the r-GO powder used for the preparation of the composite aerogels exhibited a value of 2 m^2^ g^−1^.

It is worth noting that even though the amount of r-GO in the composite aerogel was 20 wt.%, the BET value of PPO_0.8_/r-GO_0.2_ was as high as 440 m^2^ g^−1^. This feature is essential, as catalyst supports must provide a considerably large surface area.

### 2.3. Catalytic Activity of the Polymer/r-GO Monolithic Aerogels

In order to assess the catalytic activity of sPS/r-GO and PPO/r-GO aerogels and the possible use of these composite aerogels for the degradation of organic compounds from water, benzyl alcohol was selected as a model compound for preliminary testing. The oxidation scheme of benzyl alcohol by GO reported in literature [42] is shown in Figure 6.

The oxidation of benzyl alcohol in the presence of GO as catalyst leads to the formation of benzaldehyde and benzoic acid, which are obtained in different relative amounts depending on the amount of GO and the reaction conditions [42].

In Figure 7, the FTIR spectra of sPS_0.8_/r-GO_0.2_ aerogels before (curve ii) and after (curve iii) immersion in a 100 ppm benzyl alcohol aqueous solution are reported. For the sake of discussion, the FTIR spectrum of benzyl alcohol is also reported (curve i).

The FTIR spectra of the sPS/r-GO aerogel before (curve ii) and after (curve iii) immersion in a 100 ppm benzyl alcohol aqueous solution presented IR bands located at 572 and 502 cm^−1^ typical of the ordered s(2/1)2 helical conformation [58], which is characteristic of the nanoporous crystalline δ-form. From Figure 7, we could also observe that the FTIR spectrum of the aerogel before (curve i) and after (curve ii) immersion in the aqueous solution of benzyl alcohol remained unchanged. In particular, none of the benzyl alcohol IR bands at 735 and 597 cm^−1^ (see curve i), which had not covered by sPS bands, were observed on the FTIR spectrum collected after immersion. This result indicated that benzyl alcohol was not absorbed from the aqueous solution by the aerogel.

In Figure 8A,B, the FTIR spectra of PPO aerogels and PPO/r-GO aerogels before (curves i) and after (curves ii) immersion in a 100 ppm benzyl alcohol solution are reported.

First of all, it is worth noting that all the FTIR spectra of both pure PPO and PPO/r-GO aerogels displayed a band located at 414 cm^−1^, which is typical of the PPO nanoporous crystalline phase [30]. The FTIR spectrum of the PPO aerogel collected after immersion in the aqueous benzyl alcohol solution (curve ii) presented a new band at 697 cm^−1^ (Figure 8A). The presence of this typical band of benzyl alcohol (see curve i of Figure 7) showed that benzyl alcohol was sorbed by the aerogel. The amount of benzyl alcohol absorbed from the aqueous solution determined by TGA was c.a. 7 wt.%. The ability of PPO to sorb benzyl alcohol, while it was not absorbed by sPS aerogels, could be attributed to the different shape and size of the PPO crystalline nanopores [27].

The FTIR spectrum of the composite PPO_0.8_/r-GO_0.2_ aerogel collected after immersion in the solution (Figure 8B) differed from the spectrum obtained with the neat PPO aerogel. In particular, with respect to curve ii of Figure 8A, a decrease in the benzyl alcohol characteristic band at 697 cm^−1^ and the appearance of a new band at 711 cm^−1^ (curve ii of Figure 8B) were observed. This result might indicate a partial conversion of benzyl alcohol.

As mentioned above (Figure 6), the oxidation of benzyl alcohol by r-GO can lead to the formation of benzaldehyde and benzoic acid. Thus, in order to determine which compound was formed in the composite PPO/r-GO aerogel, the FTIR spectra of neat PPO aerogel after sorption of benzaldehyde and benzoic acid from 100 ppm aqueous solutions were collected (Figure 8C).

After sorption of benzaldehyde, the FTIR spectrum of the PPO aerogel displayed strong absorbance bands located at 689 and 650 cm^−1^ (curve iii of Figure 8C), while the appearance of a strong band located at 711 cm^−1^ (curve ii of Figure 8C) was observed after sorption of benzoic acid. This result suggested that the oxidation of sorbed benzyl alcohol in the PPO/r-GO aerogel led to the formation of benzoic acid. Based on the decrease in the absorbance of the 697 cm^−1^ band typical of benzyl alcohol, the fraction of benzyl alcohol being oxidized could be estimated as c.a. 50%.

## 3. Materials and Methods

### 3.1. Materials

The high molecular weight PPO (Ultra High P6130 grade; M_w_ = 350 kg mol^−1^) and sPS (90ZC grade; M_w_ = 200 kg mol^−1^) used in this study were kindly supplied by Sabic (Bergen op Zoom, The Netherlands) and Idemitsu Kosan Co., Ltd. (Tokyo, Japan), respectively. Graphite, with an 8427 trademark, was purchased from Asbury Graphite Mills Inc. (Asbury, OH, USA) Solvents and other reagents used in this work were purchased from Sigma Aldrich (St. Louis, MO, USA) and used as received.

### 3.2. Graphene Oxide and Reduced Graphene Oxide Synthesis

Hummers’ method was adapted to the synthesis of graphite oxide (GO) [59]. Briefly, 120 mL of H_2_SO_4_ and 2.5 g of NaNO_3_ were introduced into a 2000 mL three-neck round-bottomed flask containing 5 g of pure graphite powder. After that, 15 g of KMnO_4_ was also added slowly with magnetic stirring; during this process, the oil bath temperature was maintained below 10 °C. Subsequently, the reaction temperature was increased to 35 °C and continued overnight. At the end of the reaction, 700 mL of deionized water and 5 mL of H_2_O_2_ (30 wt.%) was added to the reaction mixture and removed from the oil bath. Additionally, 7 L of deionized water was added and then centrifuged at 10,000 rpm for 15 min with a Hermle Z 323 K centrifuge. Then, the isolated GO powders were repeatedly washed with a 5 wt.% HCl aqueous solution and later with deionized water. Finally, the GO powders were collected and dried at 60 °C for 12 h.

The r-GO dispersions were obtained by solvothermal reduction treatment by the addition of the appropriate amount of GO in 5 mL of *O*-dichlorobenzene, followed by sonication treatment in a 5000 mL batch bath sonicator (Badelin Sonorex RK 1028 H) at 100 °C for 120 min. Then, powder was recovered by filtration followed by *O*-dichlorobenzene extraction with scCO_2_.

### 3.3. Monolithic Polymer Aerogels Fabrication

Two different solvent systems were used in gel preparation with respect to the polymer type. For example, sPS polymer gels were prepared with *O*-dichlorobenzene, whereas the PPO polymer gels were prepared with CCl_4_ solvent. 

Gels were prepared in hermetically sealed test tubes by heating the mixture until complete dissolution of the polymer and obtaining a homogeneous solution. The hot solution was cooled down to room temperature, where gelation occurred. Subsequently, the prepared gels were extracted with a SFX 200 supercritical carbon dioxide extractor (ISCO Inc., Lincoln, NE, USA), using the following conditions: temperature (T), 40 °C; pressure (P), 250 bar; extraction time duration, 5 h.

The corresponding polymer/r-GO nanocomposite gels (sPS_0.8_/r-GO_0.2_ and PPO_0.8_/r-GO_0.2_) with 20 wt.% of r-GO loading were also prepared. Here, we used the polymer and r-GO, including the concentration as 10 wt.% of the total solvent weight (C_pol_ = 10%).

The method for polymer/r-GO nanocomposite gel preparation and solvent extraction was the same as that followed for the neat polymer gels, except for one condition. Before complete dissolution of the polymer at high temperature, the GO dispersed solution was initially sonicated for two hours at 100 °C.

### 3.4. Characterization

For samples of monolithic aerogels with a regular cylindrical shape, the total porosity, including macroporosity, mesoporosity, and microporosity, can be estimated from the volume/mass ratio of the aerogel.

Then, the percentage of porosity (*P*) of the aerogel samples can be expressed as follows [60]:(1)$$P=100\left(1-\frac{\rho_{app}}{\rho_{S}}\right)$$
where *ρ_s_* is the aerogel skeletal density (i.e., polymer matrix or polymer/r-GO matrix) and *ρ_app_* is the aerogel apparent density calculated from the mass/volume ratio of the monolithic aerogels.

X-ray diffraction measurements were performed with a Bruker D8 automatic diffractometer (Billica, MA, USA). It was operated at a step size of 0.03°, at a rate of 164 s/step, and with nickel-filtered Cu Kα radiation. Fourier transform infrared (FTIR) spectra were obtained from the Vertex 70 Bruker spectrometer equipped with a deuterated triglycine sulfate (DTGS) detector and a Ge/KBr beam splitter. The frequency scale was internally calibrated to 0.01 cm^−1^ using He-Ne laser. All measurements were made at 2.0 cm^−1^ resolution and 32 scans to reduce the noise level.

The morphology of the monolithic aerogels was characterized by a scanning electron microscope (SEM, Zeiss Evo50 equipped with an Oxford energy dispersive X-ray detector, Oxford, UK), with a low energy electron beam (5 keV). In order to avoid the surface charge and to improve the image resolution, the samples were gold-coated (~20 nm), by VCR high resolution indirect ion-beam sputtering system, before imaging.

Surface area and pore volumes of monolithic polymer aerogels were obtained by N_2_ adsorption measurements carried out at 77 K on a Nova Quantachrome 4200e sorption analyzer (Odelzhausen, Germany). The specific surface area of the aerogels was calculated using the Brunauer-Emmet-Teller method. The mesopore volumes and average radii were calculated by the Barrett-Joyner-Halenda (BJH) method using the desorption branch, while micropore cumulative volume was evaluated using the DFT method.

The sorption and oxidation tests of benzyl alcohol from aqueous solutions with sPS and PPO composite aerogels were conducted using thin pieces of aerogel (thickness, ~0.5 mm; weight, ~5 mg) soaked in a 200 mL solution. Benzyl alcohol uptake from diluted aqueous solutions was quantified by thermogravimetric measurements (TGA), using TG 209 F1 from Netzsch Instruments (Selb, Germany). All measurements were performed under controlled nitrogen gas flow and a heating rate of 10 °C/min.

## 4. Conclusions

SPS/r-GO and PPO/r-GO thermoreversible gels can be easily prepared by dissolution of sPS and PPO in r-GO dispersions in organic solvents. After the extraction of the solvents from native gels with supercritical carbon dioxide, high-porosity monolithic composite aerogels can readily be obtained. These aerogels are characterized by a fibrillar morphology, which ensures good handling properties and a homogeneous dispersion of r-GO within the fiber network.

The X-ray diffraction analysis showed that nanoporous crystalline phases were obtained within the aerogels and this led to the high BET values for the sPS-based aerogels (290 m^2^ g^−1^) and PPO-based aerogels (440 m^2^ g^−1^).

The oxidation tests carried out with dilute aqueous solutions of benzyl alcohol clearly showed that, unlike sPS-based aerogels, PPO-based aerogels were capable of absorbing benzyl alcohol from the dilute solutions and oxidation of c.a. 50% of the sorbed benzyl alcohol molecules into benzoic acid occurred.

## Figures and Tables

**Figure 1 molecules-25-05241-f001:**
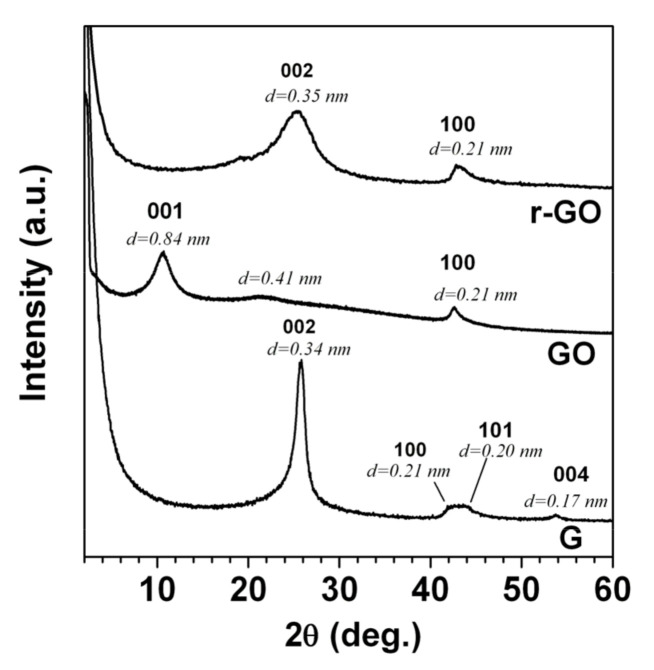
X-ray diffraction patterns of graphite (G), the resultant graphene oxide (GO) obtained by the Hummers’ method, and the final reduced graphene oxide (r-GO) obtained by solvothermal reduction.

**Figure 2 molecules-25-05241-f002:**
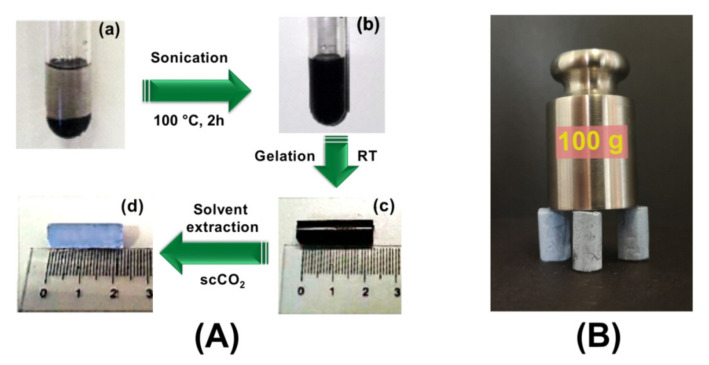
(**A**) Preparation procedure for monolithic polymer/r-GO aerogel: dispersion of GO into an organic solvent before (**a**) and after (**b**) sonication at 100 °C for 2 h in step one; (**c**) stable self-standing gel formed after adding the polymer to the dispersion of GO and the dissolution of the polymer at high temperature; (**d**) monolithic aerogel of polymer/r-GO nanocomposites fabricated by supercritical carbon dioxide (scCO_2_) extraction of gels obtained in step two. (**B**) Three syndiotactic polystyrene (sPS)/r-GO aerogel pillars (porosity = 90%) supporting 100 g.

**Figure 3 molecules-25-05241-f003:**
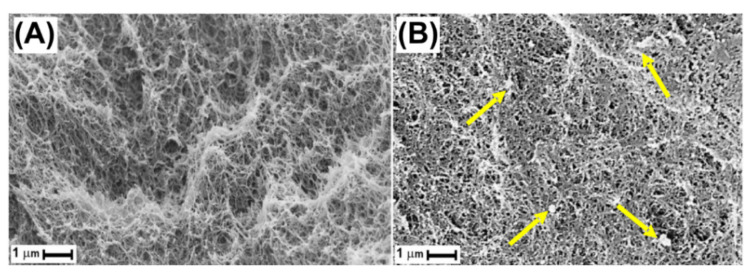
SEM images of the morphology of the monolithic pristine sPS aerogel (**A**) and of the hybrid sPS/r-GO aerogel obtained after scCO_2_ solvent extraction (**B**). Some r-GO particles dispersed in the hybrid aerogel are indicated by yellow arrows.

**Figure 4 molecules-25-05241-f004:**
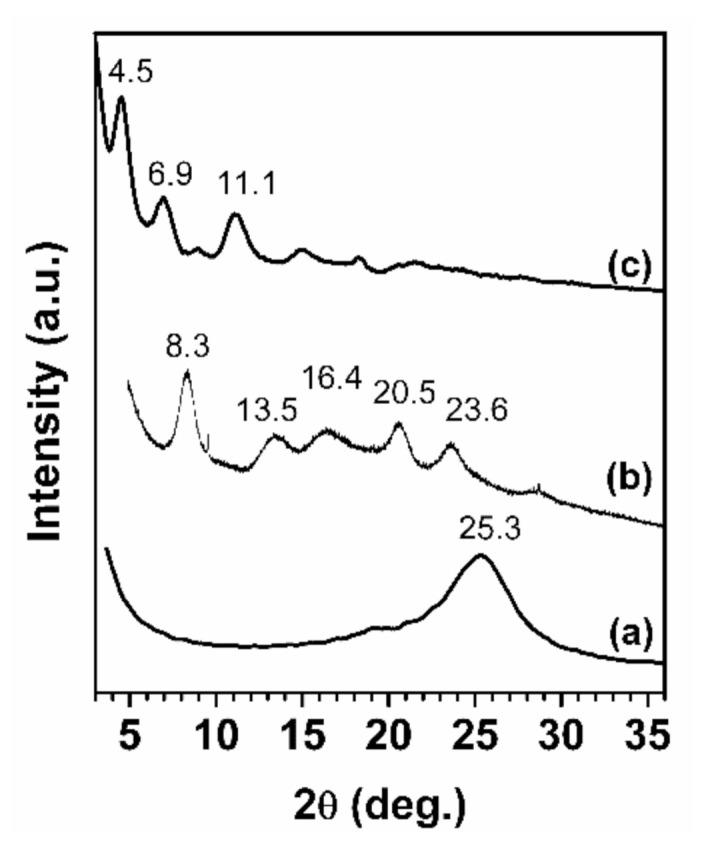
X-ray diffraction patterns of r-GO (a) and hybrid monolithic polymer aerogels of sPS_0.8_/r-GO_0.2_ (b), and PPO_0.8_/r-GO_0.2_ (c) nanocomposites containing 20 wt.% of r-GO.

**Figure 5 molecules-25-05241-f005:**
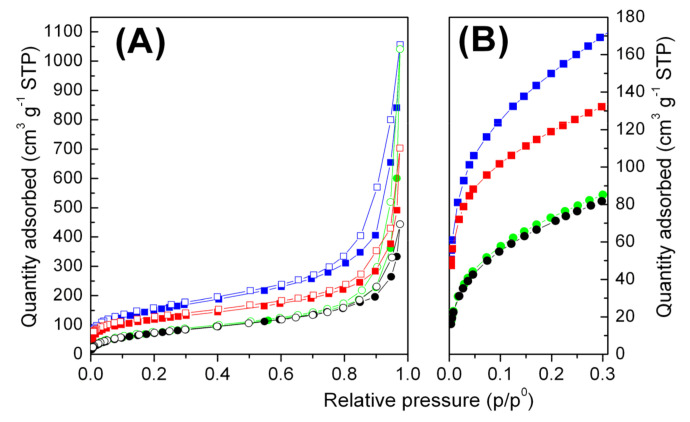
Volumetric N_2_ adsorption–desorption isotherms at 77 K in the 0–1 P/P_0_ range (**A**) and in the 0–0.3 P/P_0_ range (**B**) of pure sPS (circles, green) and PPO (squares, blue) aerogels and of sPS/r-GO (circles, black) and PPO/r-GO (squares, red) nanocomposites aerogels containing 20 wt.% of r-GO. Filled and empty scatters refer to the adsorption and desorption branches, respectively.

**Figure 6 molecules-25-05241-f006:**
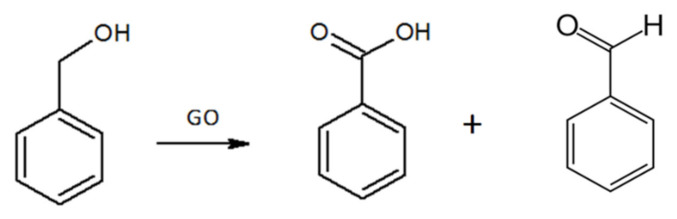
Reaction scheme shows benzyl alcohol oxidation in the presence of GO catalyst.

**Figure 7 molecules-25-05241-f007:**
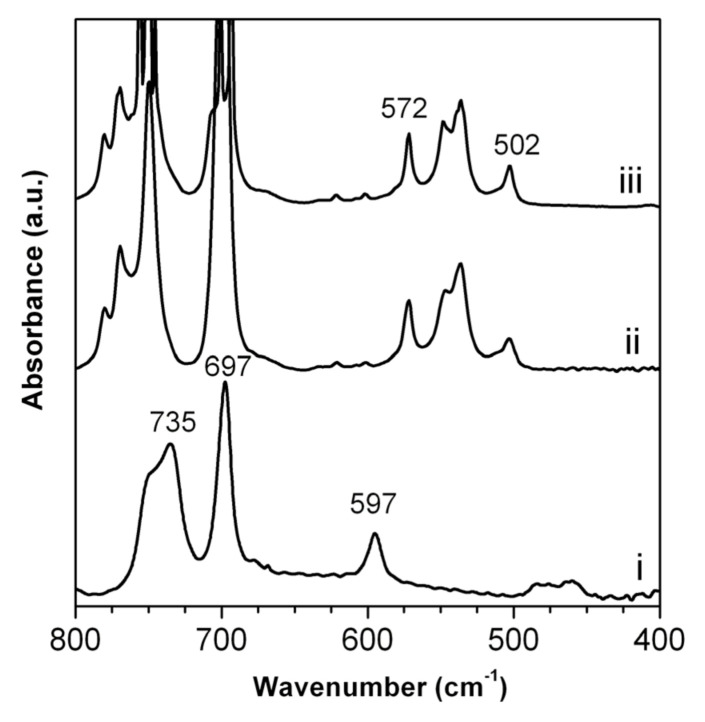
FTIR spectra of liquid benzyl alcohol (curve i) and sPS_0.8_/r-GO_0.2_ aerogel (P = 90%) before (curve ii) and after (curve iii) immersion for 3 days in a 100 ppm benzyl alcohol aqueous solution.

**Figure 8 molecules-25-05241-f008:**
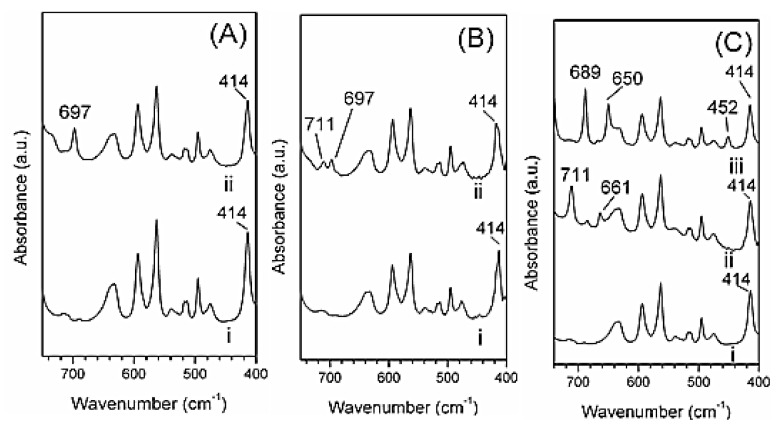
(**A**) FTIR spectra of PPO aerogels before (curve i) and after (curve ii) immersion in a 100 ppm aqueous solution of benzyl alcohol; (**B**) FTIR spectra of PPO_0.8_/r-GO_0.2_ aerogels before (curve i) and after (curve ii) immersion for 4 days in a 100 ppm benzyl alcohol solution; (**C**) FTIR spectra of PPO aerogels before (curve i) and after immersion in a 100 ppm solution of benzoic acid (curve ii) and benzaldehyde (curve iii).

**Table 1 molecules-25-05241-t001:** Summary of the apparent density (g cm^−3^), total porosity (cm^3^ g^−1^), pore volume (cm^3^ g^−1^), mesopore radius (nm), and surface area (S_BET_, m^2^ g^−1^) of the pure polymer aerogels and hybrid polymer aerogels with 10 wt.% polymer of the total solvent weight and 80/20 polymer/r-GO weight fraction.

	sPS	sPS_0.8_/r-GO_0.2_	PPO	PPO_0.8_/r-GO_0.2_
Apparent density (g cm^−3^)	0.15	0.16	0.18	0.20
Total porosity ^a^ (cm^3^ g^−1^/%)	6.66/86	6.25/80	5.55/82	5/76
S_BET_ ^b^ (m^2^/g)	290 ± 20	250 ± 20	550 ± 10	440 ± 10
V_tot_N_2_ ^c^ (cm^3^ g^−1^)	0.56	0.41	1.013	0.628
V_meso_ ^d^ (cm^3^ g^−1^)	0.49	0.33	0.86	0.48
Average mesopore radius ^d^ (nm)	9.0	6.3	8.9	8.7
V_micro_ ^e^ (cm^3^ g^−1^)	0.049	0.046	0.13	0.12

^a^ Total porosity estimated from the volume/mass ratio expressed as cm^3^·g^−1^ and % using Equation (1). ^b^ Total area evaluated following the BET model. ^c^ Pore volume (V_tot_N_2_) calculated as the volume of N_2_ liquid at P/P_0_ ≈ 0.95. ^d^ Average mesopore radius and V_meso_ calculated with the BJH method. ^e^ Cumulative micropore volume obtained from DFT analysis. PPO, poly(2,6-dimethyl-1,4-phenylene oxide).

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
