# Peer review of "Nanoporous Crystalline Composite Aerogels with Reduced Graphene Oxide"

_molecules, 2020, doi:10.3390/molecules25225241_

Round 1
Reviewer 1 Report
In present paper, authors fabricated the monolithic aerogels based on nanoporous-crystalline PPO and sPS polymers and r-GO hybrid nanocomposites. The uniform dispersion of r-GO was controlled in a two steps approach, gelation followed by scCO2 extraction process. The present aerogels shows fibrillary morphology which ensures good mechanical properties and large surface area.
The present manuscript is systematically arranged. I recommend it for publication after minor revision.
1. Abstract needs to be improved. Please revise it.
2. Author should revise the keywords.
3.Author should cite some recent references related to aerogels.
J. Alloys Compd. 805 (2019) 120-129, Journal of Alloys and Compounds 823 (2020) 153847, Nanomaterials 9 (3) (2019) 358,
4. Provide the Raman spectra for the prepared aerogel samples.
Reviewer 2 Report
The authors inform about the preparation and catalytic behaviour of crystalline composite aerogels with reduced graphene oxide.
They present a clear description of the composite synthesis that I consider perfectly reproducible. The characterization techniques are the adequate ones.
The authors claim that the composite are nanoporous. I see that nanopores exists but my question is at which extent their presence is high enough to consider the product as nanoporous when most of their porous volume corresponds to macroporosity. I am missing the corresponding pore size distribution graphs for a sounder discussion.
I do not completely understand what "nanoporous-crystalline polymer" means.
I find the catalytic activity test correct and their result quite interesting
I consider excessive to include a 25% of self-citations on the total cites. Probably, some of these cited paper contain cited of some other o this list.
The unit of the scale of absolute temperature is the kelvin and its symbol is K, not ºK.
In the line 316, page 10, there is a typing error
